# How Supplementary or Night-Interrupting Low-Intensity Blue Light Affects the Flower Induction in Chrysanthemum, a Qualitative Short-Day Plant

**DOI:** 10.3390/plants9121694

**Published:** 2020-12-02

**Authors:** Yoo Gyeong Park, Byoung Ryong Jeong

**Affiliations:** 1Institute of Agriculture and Life Science, Gyeongsang National University, Jinju 52828, Korea; ygpark615@gmail.com; 2Division of Applied Life Science (BK21 Plus Program), Graduate School, Gyeongsang National University, Jinju 52828, Korea; 3Research Institute of Life Science, Gyeongsang National University, Jinju 52828, Korea

**Keywords:** blue LED, flower bud formation, number of flowers, photoperiod

## Abstract

This research examined the effects of the supplementary or night-interrupting (NI) blue (B) light supplied at a low intensity on the flowering, gene expression, and morphogenesis of chrysanthemum, a qualitative short-day plant. White (W) light-emitting diodes (LEDs) were used to provide light with a photosynthetic photon flux density (PPFD) of 180 μmol·m^−2^·s^−1^ during the photoperiod to grow the plants in a plant factory. The control group was constructed with plants that were exposed to a 10-h short day (SD10) treatment without any blue light. The B light in this research was used for 4 h to either (1) extend the photoperiod for plants at the end of a 9-h short day (SD) treatment as the sole light source (SD9 + 4B), (2) provide night interruption (NI) to plants in the 13-h long-day (LD) treatment (LD13 + NI − 4B), (3) provide NI to plants in the 10-h SD treatment (SD10 + NI − 4B), or (4) supplement the W LEDs at the end of a 13-h LD treatment (LD13 + 4B). Blue LEDs were used to provide the supplementary/NI light at 10 μmol·m^−2^·s^−1^ PPFD. The LD13 + NI − 4B treatment resulted in the greatest plant height, followed by LD13 + 4B. Plants in all treatments flowered. It is noteworthy that despite the fact that chrysanthemum is a qualitative SD plant, chrysanthemum plants flowered when grown in the LD13 + 4B and LD13 + NI − 4B treatments. Plants grown in the LD13 + 4B had the greatest number of flowers. Plants grown in the LD13 + 4B treatment had the highest expression levels of the *cryptochrome 1*, *phytochrome A*, and *phytochrome B* genes. The results of this study indicate that a 4-h supplementation of B light during the photoperiod increases flower bud formation and promotes flowering, and presents a possibility as an alternative method to using blackout curtains in LD seasons to practically induce flowering. The B light application methods to induce flowering in SD plants requires further research.

## 1. Introduction

Plants adapt to the signals, such as the light quality, they perceive from the environment and accordingly modify their biological cycles [1]. Different types of photoreceptors, such as cryptochromes and phytochromes, enable plants to perceive changes in the light quality [2,3]. Throughout their lifecycle, the growth and development of plants are influenced by the photoreceptors. Photoreceptors monitor the light environment and also help plants time key developmental transitions, such as flowering and seed germination [4]. Phytochrome is a photoreceptor that primarily absorbs red (R) and far-red (Fr) lights, while cryptochrome is a photoreceptor that primarily absorbs ultraviolet-A (UV-A) and blue (B) lights, both of which help regulate flowering [5]. Multiple cryptochrome (*CRY1* and *CRY2*) and phytochrome (*PHYA*, *PHYB*, *PHYC*, *PHYD*, and *PHYE*) varieties can exist, depending on the species [6,7].

Light supplementation is often utilized for enhancing the quality of seedlings and rooted cuttings [8]. Photoperiod manipulation can reduce the production time and improve the overall crop quality to reduce production costs [9]. Light supplementation may take the form of supplementary light in a background of natural light, or additional light that extends the day length [8]. Night interruption (NI) interrupts a length of dark period with lighting, thus creating modified long-day (LD) conditions [10,11].

Studies have reported that B light negatively affects stem elongation and leads to a reduced leaf area [12,13,14,15,16]. Senger [17] found that blue light played a pivotal role in chloroplast development and formation, as well as the stomatal opening. It has been suggested that photoreceptors related to B light played a part in the flowering process [18,19]. Jeong et al. [20] reported that supplementary blue light at least in part promotes the elongation of stems and internodes without inhibiting the flower bud formation. In the short-day (SD) plant chrysanthemum, NI with B light did not effectively inhibit flowering, although B light is part of visible light [21,22]. Our previous study [11] split the traditional 4-h NI into two 2-h periods and shifted the NI light quality to examine how these changes affect the flowering and morphogenesis of chrysanthemum. They found out that B, Fr, R, and white (W) lights used in the first 2 h of the NI did not affect the morphogenesis nor flowering, while the same lights used in the last 2 h of the NI significantly impacted the morphogenesis and flowering. In addition, they discovered that flowering was induced in all NI treatments concluding with a blue light. Hence, we hypothesized that blue light at a low intensity supplemented to either LD or SD conditions may induce flowering in SD plants. Therefore, this study examined the effects of low-intensity (10 μmol m^−2^ s^−1^ PPFD) blue light used as supplementary or NI light on the flowering, gene expression, and morphogenesis in chrysanthemum ‘Gaya Yellow’ (a qualitative SD plant).

## 2. Materials and Methods

### 2.1. Growth Conditions and Plant Materials

Chrysanthemum (*Dendranthema grandiflorum* ‘Gaya Yellow’) spray-type cuttings were stuck in plug trays with 50 cells each filled with a commercial Tosilee Medium (Shinan Grow Company, Jinju, Korea). The cuttings were subsequently put on a glasshouse bench to root. The cuttings were relocated 12 days after they were stuck, when they have rooted, to a closed walk-in growth chamber that is 7700 cm by 2500 cm by 2695 cm in size. There, the plants were acclimatized to 20 ± 1 °C, 60% ± 10% RH, and 140 μmol·m^−2^·s^−1^ PPFD supplied with F48T12-CW-VHO fluorescent lamps (Philips Co., Ltd., Eindhoven, The Netherlands). The closed walk-in growth chamber was constructed such that numerous uniformly distributed holes allowed conditioned air to blow horizontally into the growing spaces. CO_2_ was supplemented from a compressed gas tank to maintain an atmospheric concentration of 350 ± 50 μmol·mol^−1^. The plants, after 11 days of acclimatization (the 16-h LD) in the growth chamber, were approximately 7.0 cm in height and were subjected to the photoperiodic light treatments. After being planted, the chrysanthemums were fertigated once a day (from 9:00 a.m. to 10:0 a.m.) throughout the experiment with a greenhouse multipurpose nutrient solution [11]. A 3-replication randomized complete block design was employed with a total of 6 plants for each treatment, with 2 plants in each replication. Within a controlled environment, the photoperiodic light treatments were randomly located in between replications to minimize the effects of the light treatment positioning.

### 2.2. Photoperiodic Light Treatments

Plants were grown with light at an intensity of 180 μmol m^−2^ s^−1^ PPFD provided by white MEF50120 LEDs (More Electronics Co. Ltd., Changwon, Korea) (Figure 1A). The different photoperiods used in this experiment, as well as the lighted period during the NI (referred to as ‘photoperiod’ hereafter) were as follows: B light with a wavelength of 450 nm was used for 4 h to either (1) extend the photoperiod at the end of a 9-h SD as the sole light source (SD9 + 4B), (2) provide NI following the 13-h LD (LD13 + NI − 4B), (3) provide NI after the 10-h SD (SD10 + NI − 4B), or (4) supplement W LEDs at the end of a 13-h LD (LD13 + 4B) (Figure 1B and Figure 2). The control was constructed by exposing the plants to a 10-h short-day treatment (SD10) without B light. B light at an intensity of 10 ± 3 μmol·m^−2^·s^−1^ PPFD was provided by LEDs for the photoperiodic light treatments. A HD2102.1 digital photometer (Delta OHM, Padova, Italy) measured the average PPFD 20 cm above the bench top, for each light treatment. The lighting was adjusted such that the same PPFD levels were provided to the plants regardless of the light treatment. A USB 2000 Fiber Optic Spectrometer (Ocean Optics Inc., Dunedin, FL, USA; detects wavelengths between 200 to 1000 nm) scanned the spectral distribution in 1-nm wavelength intervals 25 cm above the bench top. 

### 2.3. Data Collection and Statistical Analysis

The dry mass, number of leaves per plant, number of nodes per plant, number of flowers per plant, plant height, leaf area, chlorophyll content, percent flowering, days of treatment needed to visible flower bud or days to visible buds (DVB), flower width, and photoreceptor gene expressions were measured after 41 days of the photoperiodic light treatments. All leaves with a length greater than 1 cm in were counted to determine the number of leaves per plant. Divided samples of the shoot and root were dried at 70 °C for 72 h in a Venticell-222 drying oven (MMM Medcenter Einrichtungen GmbH., Munich, Germany) before the dry mass measurements were taken with an EW 220-3NM electronic scale (Kern and Sohn GmbH., Balingen, Germany). Leaf area measurements were taken with a LI-3000 leaf area meter (LI-COR Inc., Lincoln, NE, USA). The chlorophyll concentration was estimated from 10-mg samples of fresh, young, and fully developed leaves. Chlorophyll was extracted with 80% acetone at 4 °C. A Biochrom Libra S22 spectrophotometer (Biochrom Co. Ltd., Holliston, MA, USA) measured the absorbance of the supernatant at 645 and 663 nm, after the extracted chlorophyll was centrifuged at 3000 rpm. Calculations were performed according to the method described by Dere et al. [23]. The statistical analysis was performed with the SAS 9.1 software (SAS Institute Inc., Cary, NC, USA). An analysis of variance (ANOVA) and Tukey’s multiple range test were performed with the results of this study. SigmaPlot 12.0 (Systat Software Inc., San Jose, CA, USA) was used for graphing.

### 2.4. Total RNA Isolation, cDNA Synthesis, and Real-Time Polymerase Chain Reaction (PCR) of Selected Genes

After 20 days of the photoperiodic light treatments, plants started displaying visible flower buds and the most recently matured 10 leaves per plant were collected for total RNA extraction. The latest leaf to be matured was collected an hour after the daily photoperiodic treatments began, at 9:00 a.m. This collection time was chosen because the photosynthetic rates are high at this time of the day. Equal amounts of cDNA using primers of *cryptochrome 1* (*CRY1*), *phytochrome A* (*PHYA*), and *phytochrome B* (*PHYB*), whose sequences are shown in Table 1, were used to perform the independent PCRs. As actin is frequently used to normalize molecular expression studies, it was used as an internal control. The 2^−ΔΔCt^ method [24] was used to determine the relative expression levels of each gene. At each sampling date, the individual gene expression levels in the plants grown with the light treatments were divided by the mean gene expression levels for plants in the control (SD10). The total RNA isolation and real-time quantitative PCR analysis of the selected genes were performed according to the method described in Park et al. [11].

## 3. Results

### 3.1. Morphogenesis

It was observed that the supplementary and night-interrupting blue light increased the plant heights in this study (Figure 3A). Plants grown in LD13 + NI − 4B had the greatest height (Figure 3A), where it was 22% greater than that of plants grown in SD10. Additionally, it was observed that even plants in SD9 + 4B had a greater mean height than those in SD10. 

The dry mass of plants grown under all photoperiodic treatments was greater compared to that of the plants in the SD10 control (Figure 3B). Increasing the photoperiod, as with LD13 + 4B and LD13 + NI − 4B, significantly increased the dry mass of the plants in this study. The other treatments, SD9 + 4B and SD10 + NI − 4B, were not as effective as the LD treatments in increasing the dry mass (Figure 3B). 

Plants in SD9 + 4B had the greatest number of leaves per plant while those in SD10 had the lowest number of leaves per plant (Figure 3C). The average leaf area was the greatest for plants in LD13 + NI − 4B and the smallest for plants in SD10 + NI − 4B (Figure 3D). The leaf area per plant was 12% for plants in SD10 + NI − 4B when compared to that for plants in SD10 (Figure 3D). Furthermore, all B light treatments except for SD10 + NI − 4B increased the leaf area compared to the control (Figure 3D). The chlorophyll levels were the lowest for plants in LD13 + 4B and the highest for plants in LD13 + NI − 4B (Figure 3E). Plants in LD13 + 4B had 32% lower chlorophyll contents than plants in SD10 did (Figure 3E).

### 3.2. Flowering and Gene Expression

The flowering percentage of plants was 100% in all treatments (Table 2 and Figure 4). The fastest flowering induction was observed for plants in the control (SD10). It is noteworthy that plants in LD13 + 4B and LD13 + NI − 4B flowered, despite the fact that chrysanthemum is a qualitative SD plant (Table 2 and Figure 4).

Plants in SD10 had the smallest DVB whereas plants in LD13 + NI − 4B had the greatest DVB (Table 2). The DVB was observed to increase as the photoperiod increased (Table 2). The DVB of plants in LD13 + 4B was smaller than those of plants in LD13 + NI − 4B. Interestingly, plants in LD13 + 4B had 93% more flowers per plant compared to plants in the SD10 control. Plants in the SD10 control and SD9 + 4B had the lowest number of flowers (Table 2). Plants in the SD10 control had the greatest flower width (Table 2). 

The photoreceptor gene expression (*PHYA*, *PHYB*, and *CRY1*) in response to the B light was also analyzed (Figure 5). Plants in LD13 + 4B had the greatest expression levels of *PHYA* and *PHYB*, followed by plants in SD10 + NI − 4B (Figure 5). *PHYA* had the lowest expression levels in plants in LD13 + NI − 4B (Figure 5). *PHYB* was the least expressed in plants in LD13 + NI − 4B (Figure 5). Plants in LD13 + NI − 4B had significantly higher *CRY1* expression levels compared to plants in the other treatments (Figure 5). *CRY1* was the least expressed in plants in LD13 + 4B (Figure 5).

## 4. Discussion

### 4.1. Plant Height

Different studies have observed that blue light acts to limit the elongation of the petiole, stem, and hypocotyl in various horticultural species, such as chrysanthemum, lettuce, pepper, and soybean [25,26,27,28,29,30,31,32]. Normally, increasing blue light decreases the stem length to a maximum threshold level [32]. Schuerger et al. [30] observed that blue light for 12 h a day plays a role in changing the stem anatomy, inhibiting the growth, and determining the morphogenetic characteristics of pepper plants. Furthermore, Khattak and Pearson [33] found that B light during the photoperiod in low-light environments resulted in reduced plant heights. Cryptochromes are also known to influence the stem elongation, and various of plants exhibit suppressed shoot elongation in response to B light in a 12-h day [26]. However, these photomorphogenic responses are different for different species. Previous studies used B light during the photoperiod to control the morphogenesis, while the current study used B light as a supplement or for NI to control morphogenesis and flowering. 

All the photoperiodic light treatments considered in this study resulted in greater plant heights than that observed in the SD10 control (Figure 3A). This indicates that blue light may be used in the production of cut chrysanthemum flowers, as longer stems are considered to be of better quality. Kong et al. [34] stated that the increased elongation growth of plants in response to B light is linked to lower phytochrome activity, and is a shade-avoidance response, where different species have different sensitivities. These results agree with those of Jeong et al. [20], where it was found that an extended photoperiod with blue light promoted stem elongation of chrysanthemum. Longer photoperiods are known to be associated with the presence of higher gibberellin levels, which enhance stem elongation in chrysanthemums [20,35]. In many species, including salvia and marigold, B light was more effective than R light in increasing the shoot elongation [36]. Muleo and Morini [37] reported that internode extension on the stem leader in apple was inhibited by B LED, which determined the lowest values among all the light qualities tested. The differing responses of different plants to B light indicates that a species’ responses to a specific light quality cannot necessarily be predicted on the basis of responses of other species.

### 4.2. Dry Mass and Leaf Growth

In this study, plants in LD13 + 4B and LD13 + NI − 4B had greater shoot and root dry masses compared to plants in the SD10 control (Figure 3B). These results indicate that a prolonged photoperiod contributed to the dry matter production. Moreover, B light supplementation increases the photosynthetic carbon assimilation and may also allow greenhouse crops to accumulate a greater biomass [38].

Plants in all the photoperiodic light treatments had a higher number of leaves per plant compared to plants in the control, and plants in SD9 + 4B had the greatest number of leaves (Figure 3C). Plants in SD10 + NI-4B had a smaller mean leaf area than plants in the SD10 control did (Figure 3D), resulting from shorter leaf lengths and widths (data not shown). All other treatments with B light led to a greater leaf area than that of plants in the SD10 control. Wang et al. [39] reported similar results, where light treatments with weak 50 μmol·m^−2^·s^−1^ PPFD B light lead to increases in the leaf area. Honecke et al. [26] proposed that B light is required during the photoperiod for the normal growth of lettuce seedlings grown under R LEDs; if the B light level was low, long, narrow leaves developed. Iacona and Muleo [40] reported that total leaf area per plant in cherry rootstock ‘Colt’ was significantly greater in B LED-exposed plants than other treatments. However, these photomorphogenic responses are specific to the particular species. Dougher and Bugbee [41] reported that increasing the B light proportion resulted in decreasing leaf area in soybean, while in lettuce, increasing the B light proportion resulted in increasing the leaf area. Eskins [42] observed that the *Arabidopsis thaliana* leaf area was negatively correlated with the B light proportion, as a high-intensity B light irradiance corresponded to the development of small leaves.

Gang et al. [43] observed that the chlorophyll levels increased as plants transitioned from the vegetative to the reproductive growth, and decreased during maturation. Correspondingly, the lower chlorophyll content of plants in LD13 + 4B compared to that of plants in the other treatments in this study (Figure 3E) may be due to the continued maturation after the plants transition from the vegetative to reproductive growth.

### 4.3. Expression of Genes Related to the Morphogenesis and Flowering 

The expression levels of *PHYA* and *PHYB* were the highest for plants in LD13 + 4B, and the expression level of *CRY1* was the greatest for plants in LD13 + NI − 4B. Plants in these two treatments were also the tallest. It has been reported that cryptochromes and phytochromes affect the height of chrysanthemums [33]. In *Arabidopsis*, high *PHYB* levels can increase the expression of *AtGAox2*, which controls the synthesis of gibberellins (GAs) [35]. Furthermore, it has been verified that both phytochromes and cryptochromes play a part in the regulation of the plant hormone GA levels [35,44]. Thus, it is speculated that the high expression levels of *PHYA*, *PHYB*, and *CRY1* found in plants grown in LD13 + 4B and LD13 + NI − 4B may promote the synthesis of GAs and eventually result in greater plant heights.

It is well known that photoreceptors related to B light were involved in the flowering process [18,20]. The CRY1 and CRY2 both mediate the flowering promotion by B light [45]. *PHYA* mediates the flowering promotion by Fr light, and *PHYB* mediates the flowering inhibition by R light in *Arabidopsis* [46,47,48]. Although *PHYA* and *PHYB* are R light receptors, it has also been shown that they also function under B light in *Arabidopsis* [49], and it has been proven that either *PHYA* or *PHYB*, as well as cryptochromes, were required for responses to B light [24,46,50]. In this study, the number of flowers per plant was shown to increase with the B light treatments. This may be attributed to the high *CRY1* expression levels. Similarly, Park et al. [11] reported that a greater number of flowers per plant was observed with light shifting from B during the NI, which may be attributed to a high light energy induction as well as shade avoidance responses, a behavior where plants evade darkness by lengthening the internodes. In rice, NI with B light delayed the flowering time, but this delay was not reproduced in the *PHYB-1* mutant [51], which means *PHYB* is a negative regulator for the flowering time. It was also observed that while chrysanthemum is a qualitative SD plant, those in the LD13 + 4B and LD13 + NI − 4B treatments still flowered. This indicates that high *PHYA* and *CRY1* expression levels may induce flowering. However, further research is necessary to verify this speculation.

In summary, B light resulted in a greater height and promoted the flowering in chrysanthemum. The results of this study illustrate that a 4-h B light supplementation during the photoperiod promoted flowering and increased the number of flower buds formed. Hence, B light supplementation may be an optimal technique to induce flowering, and can be practically applied to commercial cultivation of SD plants. This study suggests that B light supplementation is an alternative practical technique to induce flowering in SD plants to using blackout curtains during LD seasons. Further research is still needed to optimize B light supplementation techniques for flowering induction of SD plants.

## Figures and Tables

**Figure 1 plants-09-01694-f001:**
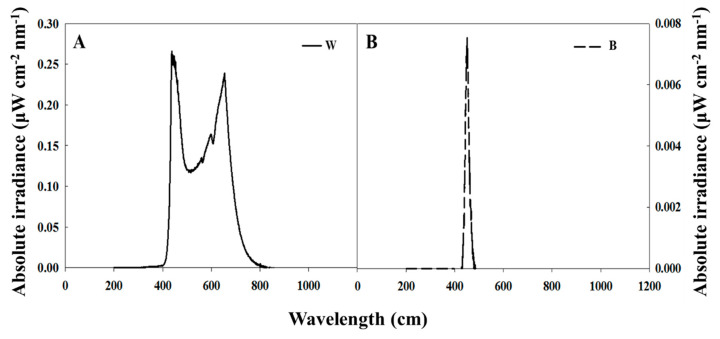
The spectral distribution of lights used in this experiment: daily W light provided by white LEDs (**A**) and B light from blue LEDs used as the supplementary and night-interrupting light (**B**).

**Figure 2 plants-09-01694-f002:**
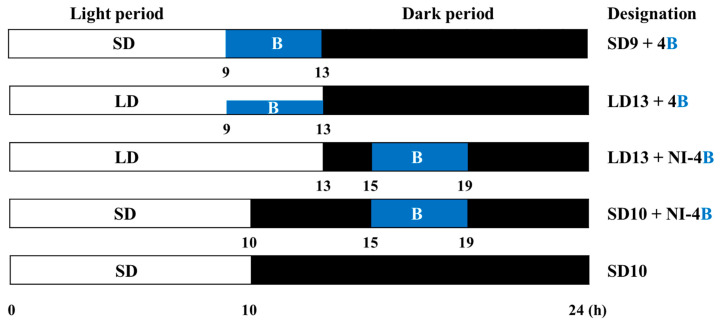
Supplementary and night-interrupting blue (B) light schemes employed in this study. B light was used for 4 h to either (1) extend the photoperiod at the end of a 9-h SD as the sole light source (SD9 + 4B), (2) provide NI following a 13-h LD (LD13 + NI − 4B), (3) provide NI after a 10-h SD (SD10 + NI − 4B), or (4) supplement the W light at the end of a 13-h LD (LD13 + 4B). Plants in the control were grown with a 10- hour SD treatment (SD10) without any B light.

**Figure 3 plants-09-01694-f003:**
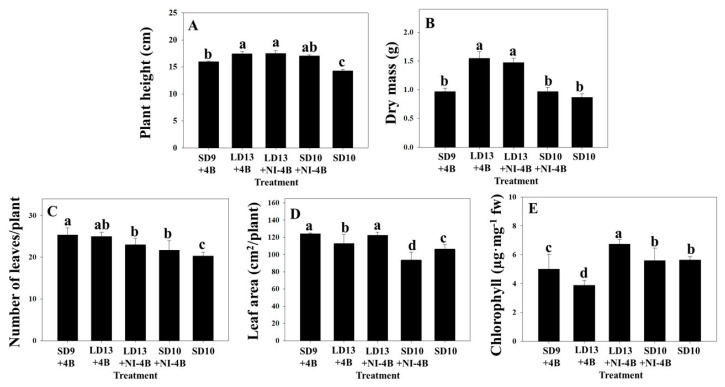
The effects of the supplementary and night-interrupting B light on the plant height (**A**), dry mass (**B**), number of leaves per plant (**C**), leaf area per plant (**D**), and chlorophyll levels (**E**) in *D**. grandiflorum* ‘Gaya Yellow’. The control was constructed by exposing plants to a 10-h SD treatment (SD10) without any B light. Data are the mean ± S.E of the 3 biological replicates. Means accompanied by different letters significantly differ (*p* < 0.05) according to Tukey’s studentized range test at a 5% significance level.

**Figure 4 plants-09-01694-f004:**
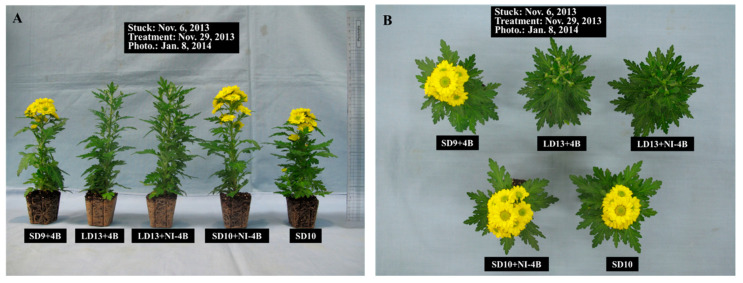
The effects of the supplementary and night-interrupting 10 μmol·m^−2^·s^−1^ PPFD B light on the flowering of chrysanthemum (*D**. grandiflorum* ‘Gaya Yellow’), after 41 days of exposure to the photoperiodic light treatments: side view (**A**) and top view (**B**) (see Figure 2 for details on the photoperiodic treatments with B light).

**Figure 5 plants-09-01694-f005:**
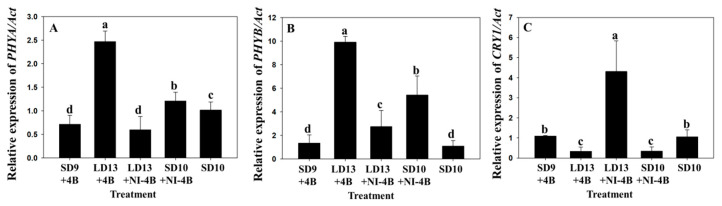
The effects of the supplementary and night-interrupting 10 μmol·m^−2^·s^−1^ PPFD B light on the relative gene expression levels of) *D**. grandiflorum* ‘Gaya Yellow’ determined by real-time PCR of *PHYA* (**A**), *PHYB* (**B**), and *CRY1* (**C**). (Details of the NI light qualities are presented in Figure 2). At each sampling date, the individual gene expression levels for the plants in the photoperiodic light treatments were divided by the mean gene expression level for plants in the SD10 control. The data are presented as the mean ± S.E of the 3 biological replicates. Means accompanied by different letters indicate significant differences (*p* < 0.05), according to Tukey’s studentized range test at a 5% significance level.

**Table 1 plants-09-01694-t001:** The primers used to quantify the gene expression levels.

Gene	Accession no.	Forward Primer	Reverse Primer
*CRY1*	NM_116961	5′-CGTAAGGGATCACCGAGTAAAG-3′	5′-CTTTTAGGTGGGAGTTGTGGAG-3′
*PHYA*	EU915082	5′-GACAGTGTCAGGCTTCAACAAG-3′	5′-ACCACCAGTGTGTGTTATCCTG-3′
*PHYB*	NM_127435	5′-GTGCTAGGGAGATTACGCTTTC-3′	5′-CCAGCTTCTGAGACTGAACAGA-3′
*Actin*	AB205087	5′-CGTTTGGATCTTGCTGGTCG-3′	5′-CAGGACATCTGAAACGCTCA-3′

**Table 2 plants-09-01694-t002:** The effects of the supplementary and night-interrupting 10 μmol·m^−2^·s^−1^ PPFD B light on the flowering characteristics of chrysanthemum (*D**. grandiflorum* ‘Gaya Yellow’), after 41 days of exposure to the photoperiodic light treatments.

Treatment ^z^	Flowering (%)	DVB ^y^ (Day)	No. of Flowers/Plant	Flower width (cm)
SD9 + 4B	100	17.7 c ^x^	11.0 c	2.6 b
LD13 + 4B	100	22.5 b	21.3 a	0.7 c
LD13 + NI − 4B	100	28.7 a	15.3 b	0.5 d
SD10 + NI − 4B	100	18.0 c	15.0 b	2.8 a
SD10	100	17.5 c	11.0 c	2.9 a
*F*-test		***	***	***

^z^ See Figure 2 for details on the photoperiodic treatments with B light. ^y^ Days of treatment to visible flower bud or days to visible buds. ^x^ Mean separation within columns by Duncan’s multiple range test at a 5% level. ***: Significant at *p* ≤ 0.001.

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
