# Peer review of "How Supplementary or Night-Interrupting Low-Intensity Blue Light Affects the Flower Induction in Chrysanthemum, A Qualitative Short-Day Plant"

_plants, 2020, doi:10.3390/plants9121694_

Round 1
Reviewer 1 Report
This study monitored the effect of dim B light on flowering and morphogenesis of a bona fide short day plant, chrysanthemum. Surprisingly, the authors found that supplement of B under LD condition will induce the flowering of chrysanthemum, though still later than SD10. The authors also examined the changes of other factors, such as plant height, dry mass, leaf area, chlorophyll etc., though the effect of B light supplement on these traits were different. This study will potentially provide conceptual guidance for the cultivation of chrysanthemum and horticulture production, which is of significance. However, I have several major concerns to be addressed:
- Please double check the statistics in Figure 3A and 3B, especially the letter labelling. In Figure 3A, there is no single column labelled as ‘b’ but there is a column labelled as ‘ab’. Similar problem in Figure 3B, there is no single column labelled as ‘b’ or ‘d’, but there is ‘ab’ and ‘cd’.
- The most striking finding of this study is that B light supplement under LD will induce flowering. The pictures shown in Figure 4 clearly showed the formation of flower buds under LD13+4B and LD13+NI-4B. But a longer period of incubation, larger than 41 days, should be shown to readers to prove that these flower buds will develop into normal flowers.
- The study is mostly involved in the investigation of flower formation and flowering. According to the study in Arabidopsis, CO and FT are two key regulators that control flowering, and multiple photoreceptor pathways converge to CO-FT pathway to regulate flowering. Thus, the expression of CO and FT under the conditions used in this study should be monitored, which, I think, is much more important than check the expression of PHYA, PHYB or CRY1. And a time course of CO and FT expression should be monitored during each photoperiodic treatment.
Author Response
1. Please double check the statistics in Figure 3A and 3B, especially the letter labelling. In Figure 3A, there is no single column labelled as ‘b’ but there is a column labelled as ‘ab’. Similar problem in Figure 3B, there is no single column labelled as ‘b’ or ‘d’, but there is ‘ab’ and ‘cd’.
Response> Thanks for the good point. As pointed out by the reviewer, the column labell was revised in Figures 3A and 3B.
2. The most striking finding of this study is that B light supplement under LD will induce flowering. The pictures shown in Figure 4 clearly showed the formation of flower buds under LD13+4B and LD13+NI-4B. But a longer period of incubation, larger than 41 days, should be shown to readers to prove that these flower buds will develop into normal flowers.
Response> Thanks for the good point. I completely agree with the reviewer. Flowers bloomed normally 41 days after treatment. Unfortunately, no related photos were left behind. I am sorry that the experiment has already ended and cannot be added.
3. The study is mostly involved in the investigation of flower formation and flowering. According to the study in Arabidopsis, CO and FT are two key regulators that control flowering, and multiple photoreceptor pathways converge to CO-FT pathway to regulate flowering. Thus, the expression of CO and FT under the conditions used in this study should be monitored, which, I think, is much more important than check the expression of PHYA, PHYB or CRY1. And a time course of CO and FT expression should be monitored during each photoperiodic treatment.
Response> The expression patterns of the CO and FT genes were not investigated. We will make sure to analyze this point by reflecting when performing the next experiment. Thank you.
Reviewer 2 Report
Authors Park and Jeong studied influence of blue light supplementation and night-interrupting on flowering and morphogenesis of chrysanthemum plants. The results in manuscript “ Flower Induction by Supplementary or Night-Interrupting, Low Intensity Blue Light in Chrysanthemum, A Qualitative Short Day Plant” is well described and discussed. I have only few comments:
- Add the average humidity in growth chamber.
- Could you specify the time of fertigation (line 85) during the photoperiod?
- Check the time information in subchapter 2.3 (line 124) and 2.4 (line 141). The expression of photoreceptor genes were measured 41 days or 20 days after initiating the photoperiodic light treatments?
Author Response
1. Add the average humidity in growth chamber.
Response> Average humidity is listed in the Materials and Methods section (lines 80-81).
2. Could you specify the time of fertigation (line 85) during the photoperiod?
Response> Authors added the time of fertigation (lines 86-87).
3.
Check the time information in subchapter 2.3 (line 124) and 2.4 (line 141). The expression of photoreceptor genes were measured 41 days or 20 days after initiating the photoperiodic light treatments?
Response> Authors checked the time information in subchapter 2.3 (line 125) and 2.4 (line 142). The expression of photoreceptor genes were measured 20 days after initiating the photoperiodic light treatments. As for the sampling time, in the case of subchapter 2.3, the growth after flowering was measured on the 41th day. In the case of subchapter 2.4, shoots had to be sampled, so growth was measured on the 20th day.
Round 2
Reviewer 1 Report
My concerns have been adequately addressed.